# Portable Wireless Intelligent Electrochemical Sensor for the Ultrasensitive Detection of Rutin Using Functionalized Black Phosphorene Nanocomposite

**DOI:** 10.3390/molecules27196603

**Published:** 2022-10-05

**Authors:** Fan Shi, Yijing Ai, Baoli Wang, Yucen Yao, Zejun Zhang, Juan Zhou, Xianghui Wang, Wei Sun

**Affiliations:** 1Key Laboratory of Water Pollution Treatment and Resource Rouse of Hainan Province, Key Laboratory of Functional Materials and Photoelectrochemistry of Haikou, College of Chemistry and Chemical Engineering, Hainan Normal University, Haikou 571158, China; 2College of Health Sciences, Hainan Technology and Business College, Haikou 570203, China; 3College of Chemical and Environmental Engineering, Chongqing University of Arts and Sciences, Chongqing 402160, China; 4College of Chemistry and Chemical Engineering, Zhaotong University, Zhaotong 657000, China

**Keywords:** portable wireless intelligent electrochemical sensor, functionalized black phosphorene nanocomposite, screen-printed electrode, rutin, electrochemistry

## Abstract

To build a portable and sensitive method for monitoring the concentration of the flavonoid rutin, a new electrochemical sensing procedure was established. By using nitrogen-doped carbonized polymer dots (N-CPDs) anchoring few-layer black phosphorene (N-CPDs@FLBP) 0D-2D heterostructure and gold nanoparticles (AuNPs) as the modifiers, a carbon ionic liquid electrode and a screen-printed electrode (SPE) were used as the substrate electrodes to construct a conventional electrochemical sensor and a portable wireless intelligent electrochemical sensor, respectively. The electrochemical behavior of rutin on the fabricated electrochemical sensors was explored in detail, with the analytical performances investigated. Due to the electroactive groups of rutin, and the specific π-π stacking and cation–π interaction between the nanocomposite with rutin, the electrochemical responses of rutin were greatly enhanced on the AuNPs/N-CPDs@FLBP-modified electrodes. Under the optimal conditions, ultra-sensitive detection of rutin could be realized on AuNPs/N-CPDs@FLBP/SPE with the detection range of 1.0 nmol L^−1^ to 220.0 μmol L^−1^ and the detection limit of 0.33 nmol L^−1^ (S/N = 3). Finally, two kinds of sensors were applied to test the real samples with satisfactory results.

## 1. Introduction

Rutin (3’,4’,5,7-tetrahydroxyflavone-3-d-rutinoside) is a typical flavonoid with various pharmacological activities such as anti-oxidant, anti-inflammatory, hypotensive, and vascular elasticity [1], which has a protective effect on the nervous system and prevents the progression of neurodegenerative diseases. It has been verified that in a variety of experimental models of nervous system diseases in vivo and in vitro, rutin is expected to be an appropriate neuroprotective agent for the treatment of ischemic stroke and other neurodegenerative diseases [2]. Therefore, it is of great significance to establish a fast and portable method for detecting the content of rutin. Different techniques have been reported for the determination of rutin such as high-performance liquid chromatography (HPLC) [3,4], chemiluminescence [5], capillary electrophoresis [6], and the electrochemical method [7,8,9,10,11]. Among them, electrochemical methods have generated great research interest over the past decade due to the merits of the fast response, high sensitivity, and minimal sample requirements. In recent years, portable wireless intelligent electrochemical sensors have been attracting tremendous attention in both fundamental studies and practical applications in various areas, including point-of-care testing (POCT), environmental monitoring, and national defense [12,13,14,15,16].

As a two-dimensional material with many unusual properties, black phosphorene (BP) exhibits a honeycomb motif similar to that of graphene [17]. However, the layers are buckled and highly anisotropic, which endows BP with many interesting properties that are distinct from those of graphene, with many future applications [18,19]. BP and BP-related composites used to improve the electrochemical sensing performance have become a research hotspot [20,21,22]. Nevertheless, because the lone-pair of electrons on its surface is strongly reactive with oxygen upon exposure to ambient conditions to form phosphate (P_x_O_y_), BP has intrinsic defects with environmental instability [23]. Therefore, our group previously developed noncovalent functionalization with a poly (3,4-ethylenedioxythio-phene)-poly (styrene sulfonic acid) hybrid film as capping layers to enhance the air stability of BP on an electrochemical hemin sensor [24]. Besides, we also designed and synthesized nitrogen-doped carbonized polymer dots (N-CPDs) anchoring few-layer BP (N-CPDs@FLBP) 0D–2D heterostructure, which was applied to build a DNA electrochemical sensor for the determination of *Escherichia coli* O157: H7 with excellent electrochemical performance and good stability [25]. The nanohybrid between FLBP and N-CPDs via the covalent interaction could resolve the aggregation of N-CPDs and environmental instability of BP with synergetic effects. N-CPDs have a polycyclic aromatic structure coated with rich hydrophilic groups, and FLBP is approximately 3 to 5 layers of single-layer BP. N-CPDs@FLBP formed a 0D-2D heterostructure that could enhance electron transport with highly ambient stability due to the formation of P-C or P-O-C bonds [26,27].

In this work, the electrochemical performances of rutin on a conventional carbon ionic liquid electrode (CILE, home-made in a laboratory) and a non-pretreated commercially available screen-printed electrode (SPE, from Qingdao Poten Technology Co., Ltd., China) modified by a N-CPDs@FLBP heterostructure and AuNPs were investigated. Here, AuNPs worked as signal amplifiers and rutin was selected as the analytical target due to its electroactive group in the structure. Finally, the proposed portable wireless intelligent electrochemical sensor and conventional CILE-based electrochemical sensor were further successfully used to determine the rutin content in certain real samples, including a rutin pharmaceutical tablet and flos sophorae immaturus (FSI). The synergetic effects of N-CPDs@FLBP and AuNPs can provide a higher conductive interface. The cation–π interaction between the aromatic imidazole rings and amine-cation of N-CPDs, the rich π electrons of FLBP and AuNPs, and the aromatic skeleton of rutin could be favorable to adsorb more rutin molecules [28]. All the above effects are beneficial to improving the performance of the rutin electrochemical sensor. Figure 1 exhibits a diagram of the electrochemical reaction mechanism of rutin on the NF/AuNPs/N-CPDs@FLBP/CILE (SPE) interface, and a comparison between the conventional CILE-based sensor and portable wireless intelligent SPE-based sensor.

## 2. Results and Discussion

### 2.1. Electrochemical Behaviors of Rutin on NF/AuNPs/N-CPDs@FLBP/CILE

Cyclic voltammograms (CV) of rutin on the different modified electrodes were surveyed in pH 3.0 PBS containing 20.0 μmol L^−1^ rutin (Figure 2a). A pair of redox peaks were observed on bare NF/CILE at 554.0 mV (E_pa_) and 509.0 mV (E_pc_). The oxidation peak current (I_pa_) is 8.77 μA, and the ratio of redox peak currents (I_pa_/I_pc_) is calculated to be 3.7 (Appendix A). Meanwhile, for NF/AuNPs/CILE and NF/N-CPDs@FLBP/CILE, visible redox peaks were observed with I_pa_ of 10.1 μA and 20.14 μA, as well as I_pa_/I_pc_ of 1.12 and 1.01, showing the sensors had better electrochemical responses towards rutin. As for NF/AuNPs/N-CPDs@FLBP/CILE, the I_pa_ (53.0 μA) is 6-fold higher than that of NF/CILE. Furthermore, the ratio of I_pa_/I_pc_ was 0.99, and the oxidation peak and the reduction peak nearly coincided, indicating that the electrochemical redox process of rutin on the surface of NF/AuNPs/N-CPDs@FLBP/CILE is more reversible [29]. The results proved that the N-CPDs@FLBP heterostructure and AuNPs can act as a more effective medium to promote the redox reaction of rutin with synergistic effects. Therefore, the better performance of rutin determination was realized on the N-CPDs@FLBP and AuNPs-based electrochemical sensor.

The electrochemical behavior of 20.0 μmol L^−1^ rutin on NF/AuNPs/N-CPDs@FLBP/CILE was evaluated in detail. The influence of PBS pH from 1.5 to 6.0 was explored by CV (Figure 2b). The oxidation peak potential (E_pa_) of rutin shows a downward trend linearly with increasing pH, which confirms the participation of protons directly in the oxidation process of rutin. On top of this, the linear equation of E_pa_ (V) = 0.657 − 0.0525 pH (R^2^ = 0.991) was obtained between E_pa_ and the pH value. As the slope (−52.5 mV pH^−1^) is close to the theoretical value (−59.0 mV pH^−1^), one can draw the conclusion that equal transfer numbers of protons and electrons are involved in the electrochemical redox of rutin, which is due to the two-electron and two-proton oxidation of the o-diphenol structure in the rutin molecule to the o-quinone structure [30]. With the increase in pH, the performance of the oxidation peak current of rutin is constantly improved and becomes worse as the pH becomes higher than 3.0. In an acidic solution, the electrochemical reaction of rutin is more likely to occur due to the presence of more protons. The highest oxidation peak current appeared at pH 3.0, which was selected as the optimal pH value.

The effects of scan rate (υ) on the CV responses were further investigated (Figure 2c). Two linear equations were obtained between the redox peak current vs. υ as I_p__a_ (μA) = −18.82 − 356.41 υ (R^2^ = 0.995) and I_p__c_ (μA) = 18.01 + 342.64 υ (R^2^ = 0.993) (Appendix A). That is to say, the redox peak currents and υ are basically proportional. Hence, it demonstrates a typical adsorption-controlled reaction process of rutin on the surface of NF/AuNPs/N-CPDs@FLBP/CILE. Furthermore, the linear relationships between the E_p_ and lnυ (Appendix A) can be used to deduce the electron transfer number (n) and the charge transfer coefficient (α) with the results as 2.1 and 0.49 by Laviron’s equation [31]. Therefore, it further manifests that there are two protons and two electrons involved in the redox reaction of rutin. The electron transfer rate constant (*k*_s_) is obtained as 4.01 s^−1^, which is larger than that of 2.39 s^−1^ on IL-CPE [29] and 3.4 s^−1^ on ERGO/GCE [32]. These results suggest that the NF/AuNPs/N-CPDs@FLBP/CILE provides fast electron transfer between the redox groups of rutin and the modified electrode.

The analytical performance for the detection of rutin on NF/AuNPs/N-CPDs@FLBP/CILE was studied by differential pulse voltammetry (DPV) (Figure 2d,e). The linear relationship of the oxidation peak current with the rutin concentration is I_pa_ (μA) = 4.30 C (μmol L^−1^) + 3.69 (*n* = 8, R^2^ = 0.990) in the range of 0.01 μmol L^−1^ to 10.0 μmol L^−1^, and I_pa_ (μA) = 0.44 C (μmol L^−1^) + 47.90 (*n* = 10, R^2^ = 0.975) in the range of 10.0 μmol L^−1^ to 180.0 μmol L^−1^ (Figure 2f). The limit of detection (LOD) is obtained as 3.3 nmol L^−1^ with a signal-to-noise (S/N) of 3. The repeatability of voltammetric responses of fifteen repetitive measurements on one electrode for voltammetric responses (Appendix A) and the reproducibility of ten electrode determinations (Appendix A) was further studied. The relative standard deviations (RSD) were calculated to be 2.93% and 5.39%, indicating high repeatability and reproducibility of the sensor for rutin detection.

To demonstrate the utility of the proposed sensor, selective studies were performed by introducing other physiological disturbance species (Figure 3a). Twenty-fold higher concentrations of rhamnose (RHA) and mannose (MAN), 50-fold higher concentrations of glucose (Glu), glycine (Gly), citric acid (CA), ascorbic acid (AA), uric acid (UA), alanine (ALA), dopamine (DA), and bisphenol (BPA), and 100-fold higher concentrations of common ions such as Na^+^, K^+^, Co^2+^, Ca^2+^, Ba^2+^, and NO_3_^−^ were examined. The initial current response was obtained for the addition of 20.0 µmol L^−1^ rutin, and the addition of the co-interfering substances to the solution did not significantly affect the DPV response for rutin (relative error, RE < 10%), which indicated the good selectivity of the proposed analytical method. However, the presence of the same concentration of quercetin (QR), catechol (CT), and resorcinol (RS) showed great influence in the rutin analysis with RE values of 45.2%, 37.2%, and 40.1%, which may be due to the similar phenol structure as rutin. Furthermore, the stability of NF/AuNPs/N-CPDs@FLBP/CILE was investigated by storing the modified electrode in a 4 °C refrigerator for 6 weeks and then used to test its performance toward rutin. The result revealed that the oxidation peak currents of rutin still remained at 81.43% of their initial values (Figure 3b), proving the good stability of the modified electrode.

### 2.2. Electrochemical Sensing of Rutin on NF/AuNPs/N-CPDs@FLBP/SPE

To further check the real application of this electrochemical sensing platform, N-CPDs@FLBP and AuNPs were applied on the surface of SPE, and further connected to a portable workstation and a smartphone. SPE is widely used due to the characteristics of simple manufacturing, a low price, easy miniaturization, and integration [16,33], and the modified electrode is flexible, meaning it can be bent by 120° (Figure 4a).

The analytical performance of rutin detection on NF/AuNPs/N-CPDs@FLBP/SPE is shown in Figure 4b–d. The linear relationships of oxidation peak currents with the rutin concentrations are I_pa_ (μA) = 6.94 C (μmol L^−1^) + 3.91 (*n* = 8, R^2^ = 0.994) from 1.0 nmol L^−1^ to 10.0 μmol L^−1^, and I_pa_ (μA) = 0.67 C (μmol L^−1^) + 81.9 (*n* = 13, R^2^ = 0.990) from 10.0 μmol L^−1^ to 220.0 μmol L^−1^ with LOD as 0.33 nmol L^−1^ (S/N = 3). The comparison of the electrochemical performance of this method with some previously reported modified electrodes for rutin detection is listed in Table 1. It can be seen that this method has a wider linear range and lower LOD. Furthermore, the portable wireless intelligent sensor shows the characteristics of high sensitivity, fast response, and portability, which have potential applications in POCT.

### 2.3. Analysis of the Practical Samples

The practical applications of NF/AuNPs/N-CPDs@FLBP/CILE (SPE) for the detection of the rutin pharmaceutical tablet and FSI samples were performed by utilizing the standard addition method with both a CILE-based sensor and an SPE-based sensor. The detection results of the two methods are listed in Table 2 with almost consistent results, which indicates that the proposed method for the intelligent detection of rutin in real samples is acceptable and applicable.

### 2.4. Possible Interaction Mechanism of Rutin and NF/AuNPs/N-CPDs@FLBP/CILE (SPE)

The probable electrochemical mechanism of rutin on NF/AuNPs/N-CPDs@FLBP/CILE (SPE) is shown in Figure 1. First of all, the aromatic skeleton of rutin may form a π–π interaction with FLBP as well as the aromatic imidazole rings of N-CPDs [28]. Hence, rutin may be absorbed on the surface of N-CPDs@FLBP through the π–π stacking interaction. Secondly, the p*K*_1_ of rutin is approximately 7, thus rutin will become positively charged in the acidic environment [41]. The positively charged rutin could easily interact with AuNPs and -NH_2_ on N-CPDs@FLBP [25] via the cation–π interaction [42]. What is more, the hydrogen bonding interaction between C–N sites of N-CPDs and the –OH groups of rutin also has a positive effect on the detection of rutin [43]. Hence, the significant sensitivity enhancement of rutin detection could be assigned to the combined effects of all these interactions, especially the π–π stacking interaction and the cation–π interaction. The larger oxidation peak current, higher sensitivity, and lower LOD of rutin detection on the SPE-based sensor may be attributed to the larger surface area with more modifiers present on the electrode interface [44]. The proposed portable electrochemical sensor can be used for the rapid and sensitive determination of trace rutin in rutin pharmaceutical tablets and medicinal plants, which show great advantages and potential applications in intelligent monitoring of rutin-related drug samples [45].

## 3. Materials and Methods

### 3.1. Materials

N-CPDs@FLBP was synthesized via a one-step microwave-assisted method based on previous work [25], which used 1-methyl-2-pyrrolidinone (NMP) and BP powder as the raw materials with microwave treatment and centrifugation to obtain the resultant nanocomposite. The detailed procedure was described in reference [25]. BP powder (>99.998 %) and gold nanoparticles (AuNPs, average particle size 40.0 nm) (Nanjing XFNANO Materials Tech Co., Ltd., Nanjing, China), NMP (Shanghai Aladdin Biochemical Technology Co., Ltd., Shanghai, China), nafion (NF, 5.0 wt% in ethanol solution, Honghaitian Tech. Co., Ltd., Beijing, China), 1-hexylpyridine hexafluorophosphate (HPPF_6_, Lanzhou Greenchem ILS. LICP. CAS., Lanzhou, China), the rutin pharmaceutical tablet (20.0 mg per tablet, Tianjin Lisheng Pharmaceutical Co., Ltd., Tianjin, China), and flos sophorae immaturus (FSI, Linshi Shengtai Pharmaceutical Co., Ltd., Haikou, China) were used in the experiment.

### 3.2. Instruments

Electrochemical measurements were conducted on a conventional CHI 660E electrochemical workstation (Shanghai CH Instrument Co., Shanghai, China) with a conventional three-electrode system, including the self-made laboratory NF/AuNPs/N-CPDs@FLBP/CILE (φ = 4.0 mm) as a working electrode, a platinum electrode as the counter electrode, and a saturated calomel electrode (SCE) as the reference electrode.

A PlamSens portable electrochemical workstation (EmStat3 + Blue, Red Matrix Co., Ltd., Guangzhou, China) was used with a modified SPE, in which PET was the substrate and a three-electrode system with a carbon disk electrode was the working electrode (φ = 5.0 mm), Ag/AgCl was the reference electrode, and a carbon ring electrode was the counter electrode (Qingdao Poten Technology Co., Ltd., Shanghai, China), to build an intelligent sensing system. By connecting a Bluetooth-enabled smartphone, the Huawei P40 Pro (Huawei Technologies Co., Ltd., Shenzhen, China), the system can transmit the electrochemical signals to the custom application on the phone.

### 3.3. Fabrication of Modified Electrodes

To fabricate N-CPDs@FLBP-modified electrodes, CILE and SPE were employed as substrate electrodes. CILE was fabricated according to the previous work [46] and polished with weighing paper before use. The modified electrode was prepared in the glove box under N_2_ protection with the following procedure: 8.0 μL of a N-CPDs@FLBP solution was first dropped on CILE and dried at 25 °C, and then 8.0 μL of a 0.2 g mL^−1^ AuNPs solution was cast and dried. Finally, 6.0 μL 0.5 wt% NF was modified on the surface, and the resultant electrode (NF/AuNPs/N-CPDs@FLBP/CILE) was used as the working electrode. For comparison, other modified electrodes, including NF/N-CPDs@FLBP/CILE, NF/AuNPs/CILE, and NF/CILE were prepared using a similar method. On the other hand, SPE can be used without any pretreatment, and the same procedure was used to obtain portable wireless intelligent electrochemical sensors to extend the practical application.

### 3.4. Samples Pretreatment

One rutin pharmaceutical tablet was finely ground and dissolved in a 10.0 mL methanol solution, which underwent ultrasound for 30 min for later analysis. The Chinese medicinal plant FSI was smashed and screened by a 100-mesh screen. Then, 0.1 g of FSI was added to 50 mL of methanol and soaked for 12 h. Then, ultrasonic extraction was performed at 55 °C for 40 min; following this, the mixture was centrifuged at 3000 r min^−1^ for 15 min to collect the supernatant, which was filtered through a 0.4 μm filter. Finally, the resulting solution was made up to 100.0 mL with methanol solution to obtain the FSI sample solution. In the analysis of the practical samples, 0.1 mL of the as-prepared rutin pharmaceutical tablet solution was added to 10.0 mL of pH 3.0 PBS to obtain three parallel samples. Then, 10.0 μL of the as-prepared FSI solution was diluted 10,000 times with pH 3.0 PBS, and then 10.0 mL of the diluent was taken for three parallel samples.

### 3.5. Electrochemical Measurements

CV was carried out in 5.0 mL 0.1 mol L^−1^ PBS containing rutin. The electrochemical performance of the samples was characterized by DPV in 0.1 mol L^−1^ PBS with DPV parameters of pulse amplitude of 0.05 V, pulse width of 0.02 s, pulse period of 0.2 s, and quiet time of 0.5 s. For modified CILE, all measurements were performed in a 5.0 mL solution, while only a 50.0 μL solution was needed for modified SPE using the portable electrochemical workstation.

## 4. Conclusions

In summary, 0D N-CPDs are hybridized with 2D FLBP to form heterostructure N-CPDs@FLBP, which is used with good stability and fast electron transport capacity due to the formation of P-C or P-O-C bonds. N-CPDs@FLBP and AuNPs were further modified on both CILE and SPE to construct electrochemical sensors for rutin detection. Due to the large surface area of SPE compared to that of CILE, more composites were modified on the SPE surface, which led to larger current responses. Furthermore, SPE exhibits certain advantages such as being portable, commercially available with less sample solution needed, and the ability to be connected to a smartphone-controlled wireless electrochemical workstation, which is more convenient for practical in situ applications. The synergistic effects of N-CPDs@FLBP and AuNPs can provide a higher conductive interface with the adsorption of rutin through the π–π stacking interaction and cation–π interaction, which are for the improvement of the electrochemical rutin-sensing ability [28,42]. These interactions led to the accumulation of rutin on the electrode surface, and better analytical performances. Finally, the feasibility is verified by comparing the results of real samples of rutin pharmaceutical tablets and Chinese medicinal plant FSI on a CILE-based sensor and an SPE-based sensor. This study provides a new idea for the rapid determination of trace rutin in Chinese medicinal plants and provides a procedure for the intelligent monitoring of the quality of Chinese medicine.

## Figures and Tables

**Figure 1 molecules-27-06603-f001:**
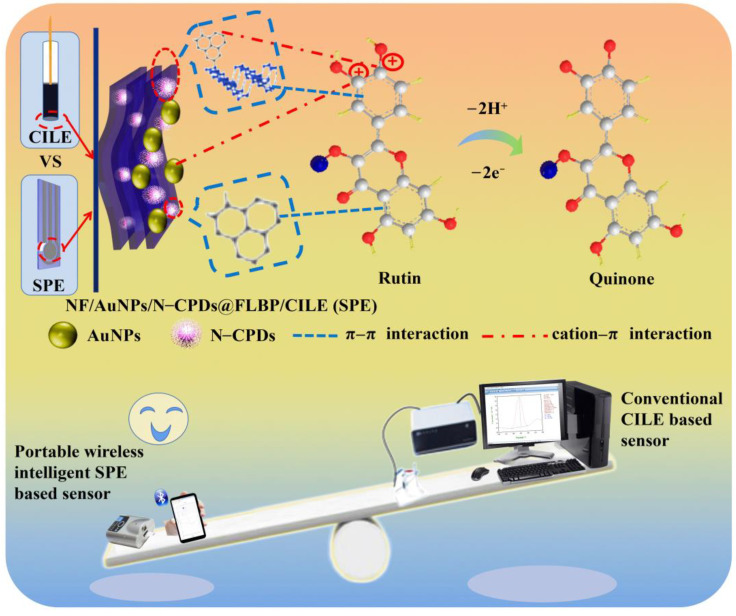
Diagram of electrochemical reaction mechanism of rutin on AuNPs/N-CPDs@FLBP/CILE (SPE) interface, and the performance comparisons of the conventional CILE-based sensor and portable wireless intelligent SPE-based sensor. (NF: nafion ethanol solution).

**Figure 2 molecules-27-06603-f002:**
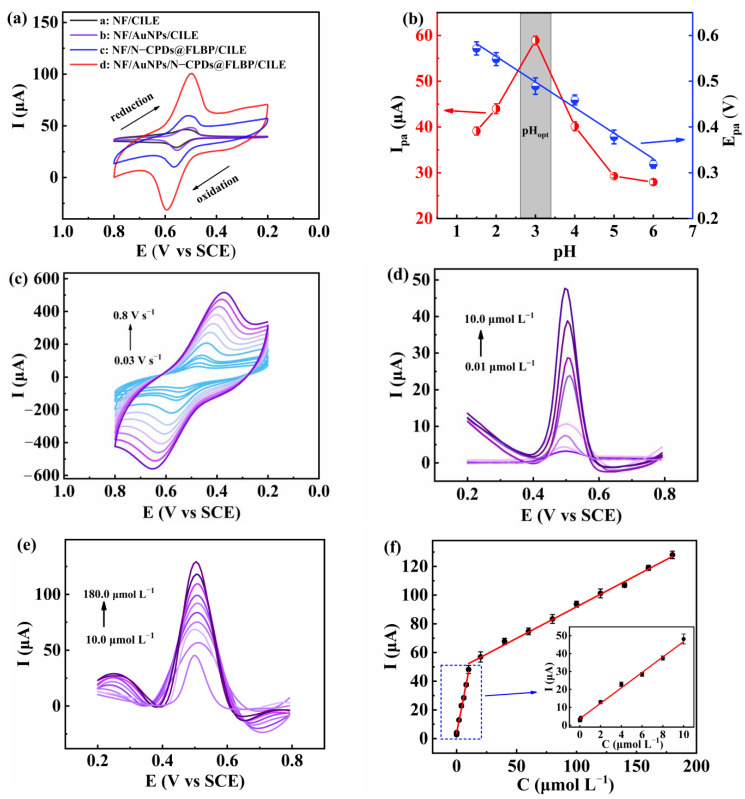
(**a**) Electrochemical behavior of 20.0 μmol L^−1^ rutin with different modified electrodes (scan rate: 0.1 V s^−1^); (**b**) the effects of different pH (1.5, 2.0, 3.0, 4.0, 5.0, 6.0) on oxidation peak potentials and oxidation peak currents for rutin on NF/AuNPs/N-CPDs@FLBP/CILE in 0.1 mol L^−1^ PBS; (**c**) CV curves of 20.0 μmol L^−1^ rutin on NF/AuNPs/N-CPDs@FLBP/CILE at different scan rates (0.03, 0.05, 0.08, 0.1, 0.2, 0.3, 0.4, 0.5, 0.6, 0.7, and 0.8 V s^−1^); DPV curves of rutin with NF/AuNPs/N-CPDs@FLBP/CILE in 0.1 mol L^−1^ PBS containing rutin concentrations from (**d**) 0.01 μmol L^−1^ to 10.0 μmol L^−1^ and (**e**) 10.0 μmol L^−1^ to 180.0 μmol L^−1^; (**f**) the linear relationship between I_pa_ and concentration of rutin (inset: The linear relationship from 0.01 μmol L^−1^ to 10.0 μmol L^−1^).

**Figure 3 molecules-27-06603-f003:**
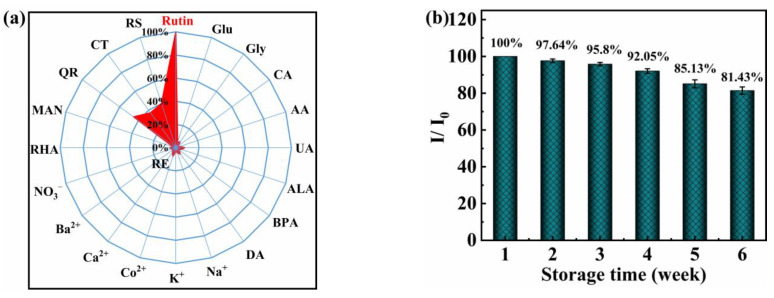
(**a**) The selectivity of NF/AuNPs/FLBP@pN-CPNDs/CILE to rutin over other analytes. The radar chart shows the RE of interferents on 20.0 μmol L^−1^ rutin determination; (**b**) oxidation peak currents of NF/AuNPs/N-CPDs@FLBP/CILE for rutin (20.0 μmol L^−1^) after storage in a 4 °C refrigerator for 6 weeks. (Notes: RHA, rhamnose; MAN, mannose; Glu, glucose; Gly, glycine; CA, citric acid; AA, ascorbic acid; UA, uric acid; ALA, alanine; DA, dopamine; BPA, bisphenol; QR, quercetin; CT, catechol; RS, resorcinol).

**Figure 4 molecules-27-06603-f004:**
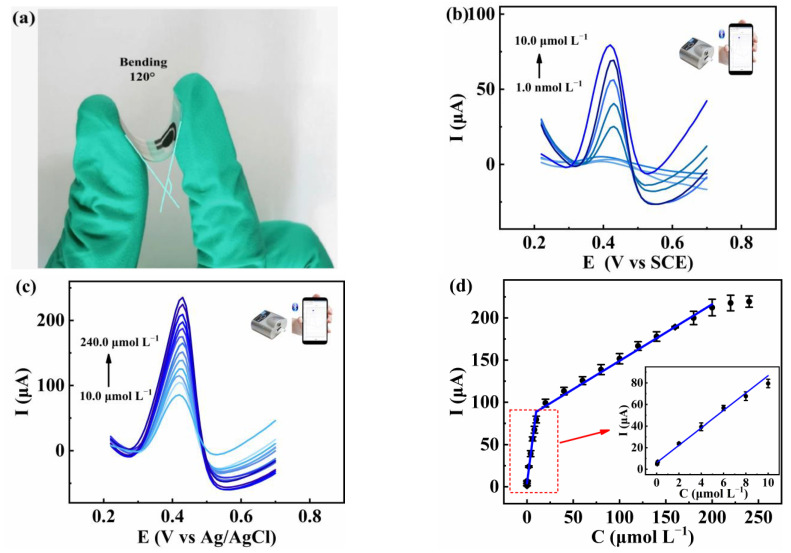
(**a**) Optical photograph of bending the modified SPE into 120°; DPV curves of rutin at NF/AuNPs/N-CPDs@FLBP/SPE in 0.1 mol L^−1^ PBS containing rutin concentrations from (**b**) 1.0 nmol L^−1^ to 10.0 μmol L^−1^ and (**c**) 10.0 μmol L^−1^ to 240.0 μmol L^−1^; (**d**) the linear relationship between I_pa_ and concentration of rutin (inset: The linear relationship from 1.0 nmol L^−1^ to 10.0 μmol L^−1^).

**Table 1 molecules-27-06603-t001:** Comparisons of the analytical performance of rutin detection with different modified electrodes.

Electrodes	Methods	Electrolytes (0.10 mol L^−1^)	Detection Ranges (μmol L^−1^)	LOD (nmol L^−1^)	Refs.
2-MBT/PGE	CV	BR (pH 4.5)	0.039–1.10, 1.10–10.50	9.60	[10]
VMSF/ErGO/ITO	DPV	PBS (pH 3.0)	0.30–2.00, 2.00–40.00	2.30	[34]
CTAC-Gr-PdNPs/GCE	SWV	PBS (pH 2.0)	0.02–1.00	5.00	[35]
AuNCs/CILE	DPV	PBS (pH 2.0)	0.004–700.00	1.33	[36]
DNA-CPIE	DPV	BR (pH 3.0)	0.008–10.00	1.30	[37]
PtNPs/RGO/GCE	DPV	PBS (pH 6.0)	0.057–102.59	20.00	[38]
BP-PEDOT: PSS/GCE	DPV	PBS (pH 6.5)	0.02–15.00, 15.00–80.00	7.00	[39]
GNR/Gr electrode	DPV	PBS (pH 7.0)	0.032–1.00	7.86	[40]
NF/AuNPs/N-CPDs@FLBP/CILE	DPV	PBS (pH 3.0)	0.01–10.00, 10.00–180.00	3.00	This work
NF/AuNPs/N-CPDs@FLBP/SPE	DPV	PBS (pH 3.0)	0.001–10.00, 10.00–220.00	0.33	This work

Notes: MBT, 2-mercaptobenzothiazole; PGE, pencil graphite electrode; BR, Britton–Robinson; VMSF, vertically ordered silica mesoporous films; ErGO, electrochemically reduced graphene oxide; ITO, indium tin oxide; CTAC, cetyltrimethylammonium chloride; Gr, graphite; GCE, glassy carbon electrode; SWV, square wave voltammetry; AuNCs, gold nanocages; CPIE, carbon paste ionic liquids electrode; RGO, reduced graphene oxide; PEDOT: PSS, poly (3,4-ethylenedioxythiophene)-poly(styrenesulfonate); GNR, graphene nanoribbon; CV, cyclic voltammograms; DPV, differential pulse voltammetry; PBS, phosphate buffer solution; LOD, limit of detection.

**Table 2 molecules-27-06603-t002:** Determinations of rutin in real samples using NF/AuNPs/N-CPDs@FLBP/CILE (SPE).

Samples	Added (μmol L^−1^)	NF/AuNPs/ N-CPDs@FLBP/CILE			NF/AuNPs/ N-CPDs@FLBP/SPE		
		Found (μmol L^−1^)	Recovery (%)	RSD (%)	Found (μmol L^−1^)	Recovery (%)	RSD (%)
Rutin pharmaceutical tablet	-	32.70	-	1.98	32.72	-	1.30
	10.0	42.51	98.10	2.05	42.98	102.80	1.12
	20.0	52.56	99.35	1.96	53.10	102.00	1.58
	30.0	63.20	101.67	1.65	62.85	100.50	1.60
FSI	-	8.62	-	2.02	8.79	-	1.94
	10.0	18.89	102.70	3.13	19.21	105.90	2.80
	20.0	28.23	95.48	2.07	28.72	100.50	2.05
	30.0	38.81	100.63	1.85	38.45	98.87	1.99

Notes: FSI, flos sophorae immaturus; RSD, relative standard deviations.

## Data Availability

Not applicable.

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
