# Peer review of "Portable Wireless Intelligent Electrochemical Sensor for the Ultrasensitive Detection of Rutin Using Functionalized Black Phosphorene Nanocomposite"

_molecules, 2022, doi:10.3390/molecules27196603_

Round 1

Reviewer 1 Report

The manuscript is related to the development of a portable wireless intelligent electrochemical sensor for the ultrasensitive detection of rutin using functionalized black phosphorene nanocomposites and testing their performance and compared with the commercial samples. The work is a thorough study in the area of novel electrochemical sensors that the authors have investigated besides an attempt to design portable and incorporate integrated wireless functions. The nanocomposites consist of a modified carbon ionic liquid electrode which they claim is the first time studied. It is a good idea not to include sentences like the first time or pioneered etc. To extend its utility, nanocomposite was further modified on the screen-printed electrode (SPEs) and used for the detection of rutin, which showed good performance with a detection range from 1.0 nM to 220.0 μM and an ultra-low detection limit of rutin (0.33 nM, S/ N 19 = 3). Finally, they studied these sensors to test real samples, and the authors obtained satisfactory results.

The paper needs re-reading and correcting some editorial and mechanical deficiencies, too numerous to mention or enlist here. Also, it will be a good idea to consolidate some of the figures or transfer them to a supplementary file. Finally, it is important that you expand the abstract to include the detection mechanism and/or in the results and discussion section, as it is not apparent. The paper is acceptable for publication with mandatory minor technical and editorial revisions.

Author Response

Response to Reviewer 1 Comments

Point 1: The paper needs re-reading and correcting some editorial and mechanical deficiencies, too numerous to mention or enlist here.

Response 1: According to the reviewer's comments, the whole paper has been carefully checked and revised to our best.

Point 2: Also, it will be a good idea to consolidate some of the figures or transfer them to a supplementary file.

Response 2: Thanks for the reviewer's comments, figure 2, figure 3 and figure 4 have been consolidated to be the new figure 2, and the others have been transferred to a supplementary file.

Point 3: Finally, it is important that you expand the abstract to include the detection mechanism and/or in the results and discussion section, as it is not apparent.

Response 3: Thanks for the reviewer's comments, the abstract has been rewritten with the detection mechanism added:

Abstract: To build a portable and sensitive method for the monitoring of the concentration of flavonoids rutin, a new electrochemical sensing procedure was established. By using nitrogen-doped carbonized polymer dots (N-CPDs) anchoring few-layer black phosphorene (N-CPDs@FLBP) 0D-2D heterostructure and gold nanoparticles (AuNPs) as the modifiers, carbon ionic liquid electrode and screen-printed electrode (SPE) were used as the substrate electrodes to construct a conventional electrochemical sensor and a portable wireless intelligent electrochemical sensor, respectively. Electrochemical behaviors of rutin on the fabricated electrochemical sensors were checked in detail with the analytical performances investigated. Due to the electroactive groups of rutin, and the specific π-π stacking and cation-π interaction between the nanocomposite with rutin, electrochemical responses of rutin were greatly enhanced on the AuNPs/N-CPDs@FLBP modified electrodes. Under the optimal conditions, an ultra-sensitive detection of rutin could be realized on AuNPs/N-CPDs@FLBP/SPE with the detection range from 1.0 nmol L-1 to 220.0 μmol L-1 and the detection limit as 0.33 nmol L-1 (S/N = 3). Finally, two kinds of sensors were applied to test the real samples with satisfactory results.

Reviewer 2 Report

This is a well-written paper on a novel technique to electrochemically detect rutin. The authors developed NF/AuNPs/N-CPDs@FLBP/CILE or SPE electrodes. All the parameters are examined carefully by the authors with systematic experiments. The scheme shows much lower limits of detection and wider detection ranges than electrods reported by other research groups. I suggest accept this paper in present form.

Typo: Page 6, Figure 5 caption first line, "pN-CNDs" should be "N-CPDs"

Author Response

Response to Reviewer 2 Comments

Point 1: Typo: Page 6, Figure 5 caption first line, "pN-CNDs" should be "N-CPDs"

Response 1: According to the reviewer's comments, "pN-CNDs" has been corrected to "N-CPDs", and the whole paper has been carefully checked and revised to our best.

Reviewer 3 Report

The manuscript entitled "Portable wireless intelligent electrochemical sensor for the ultrasensitive detection of rutin using functionalized black phosphorene nanocomposite" is interesting article and very useful for researchers. However, the minor comments are noticed. 

(i)Rewrite the abstract in the order of (a) the overall purpose of the study(b) the basic design of the study(c)major findings.

(ii) Add more significance of Functionalized black phosphorene nanocomposite in introduction section of the manuscript. 

(iii) What are the main difference between the electrochemical performance of rutin on a conventional carbon ionic liquid electrode (CILE, home-made in laboratory) and non-pretreated commercially available screen-printed electrode (SPE, from Qingdao Poten Technology Co., Ltd., China).

(iv) "The proposed portable electrochemical sensor can be used for the rapid and sensitive determination of trace rutin in rutin pharmaceutical tablet and medicinal plants, which show great advantage and application potential in intelligent monitoring the quality of Chinese medicine"Provide citation. 

(v) On line 213 of page 9, N-CPDs@FLBP was synthesized via one-step microwave-assisted method based on previous work. Make it clear of the previous work.

(vi) Rewrite conclusion by adding previous literature which supports recent findings. 

(vii) Correction on English language is necessary throughout the manuscript. 

Author Response

Response to Reviewer 3 Comments

Point 1: Rewrite the abstract in the order of (a) the overall purpose of the study (b) the basic design of the study (c) major findings.

Response 1: According to the reviewer's comments, the abstract has been rewritten in the mentioned order.

Abstract: To build a portable and sensitive method for the monitoring of the concentration of flavonoids rutin, a new electrochemical sensing procedure was established. By using nitrogen-doped carbonized polymer dots (N-CPDs) anchoring few-layer black phosphorene (N-CPDs@FLBP) 0D-2D heterostructure and gold nanoparticles (AuNPs) as the modifiers, carbon ionic liquid electrode and screen-printed electrode (SPE) were used as the substrate electrodes to construct a conventional electrochemical sensor and a portable wireless intelligent electrochemical sensor, respectively. Electrochemical behaviors of rutin on the fabricated electrochemical sensors were checked in detail with the analytical performances investigated. Due to the electroactive groups of rutin, and the specific π-π stacking and cation-π interaction between the nanocomposite with rutin, electrochemical responses of rutin were greatly enhanced on the AuNPs/N-CPDs@FLBP modified electrodes. Under the optimal conditions, an ultra-sensitive detection of rutin could be realized on AuNPs/N-CPDs@FLBP/SPE with the detection range from 1.0 nmol L-1 to 220.0 μmol L-1 and the detection limit as 0.33 nmol L-1 (S/N = 3). Finally, two kinds of sensors were applied to test the real samples with satisfactory results.

Point 2: Add more significance of functionalized black phosphorene nanocomposite in introduction section of the manuscript.

Response 2: Thanks for the reviewer's comments, more significance of functionalized black phosphorene nanocomposite has been added in introduction section: N-CPDs have polycyclic aromatic structure coated with rich hydrophilic groups and FLBP is approximately 3 to 5 layers of single layer BP. N-CPDs@FLBP formed a 0D-2D heterostructure that could enhance the electron transport with highly ambient-stability due to the formation of P-C or P-O-C bonds [26, 27].

Point 3: What are the main difference between the electrochemical performance of rutin on a conventional carbon ionic liquid electrode (CILE, home-made in laboratory) and non-pretreated commercially available screen-printed electrode (SPE, from Qingdao Poten Technology Co., Ltd., China).

Response 3: Thanks for the reviewer's comments, the main differences between the electrochemical performance of rutin on a conventional carbon ionic liquid electrode (CILE, home-made in laboratory) and non-pretreated commercially available screen-printed electrode (SPE, from Qingdao Poten Technology Co., Ltd., China) have been discussed and added in the “Conclusion” section: Due to the large surface area of SPE than that of CILE, more composites were modified on SPE surface, which lead to the larger current responses. Also SPE exhibits the advantages such as portable, commercially available with less sample solution needed, and can be connected with wireless electrochemical workstation with smartphone controlled, which are more convenient for the practical in-situ application.

Point 4: "The proposed portable electrochemical sensor can be used for the rapid and sensitive determination of trace rutin in rutin pharmaceutical tablet and medicinal plants, which show great advantage and application potential in intelligent monitoring the quality of Chinese medicine"Provide citation.

Response 4: According to the reviewer's comments, the citation was added as reference [46].

Point 5: On line 213 of page 9, N-CPDs@FLBP was synthesized via one-step microwave-assisted method based on previous work. Make it clear of the previous work.

Response 5: According to the reviewer's comments, N-CPDs@FLBP was synthesized via one-step microwave-assisted method based on previous work, which used 1-methyl-2-pyrrolidinone and BP powder as the raw materials with microwave treatment and centrifugation to obtain the resultant nanocomposite. Detailed procedure was described in the reference [25].

Point 6: Rewrite conclusion by adding previous literature which supports recent findings.

Response 6: According to the reviewer's comments, conclusion was rewritten by the addition of more discussions and previous literatures.

Sum up, 0D N-CPDs are hybridized with 2D FLBP to form heterostructure N-CPDs@FLBP, which is used with good stability and fast electron transport capacity due to the formation of P-C or P-O-C bonds. N-CPDs@FLBP and AuNPs were further modified on both CILE and SPE to construct electrochemical sensors for rutin detection. Due to the large surface area of SPE than that of CILE, more composites were modified on SPE surface, which lead to the larger current responses. Also SPE exhibits the advantages such as portable, commercially available with less sample solution needed, and can be connected with wireless electrochemical workstation with smartphone controlled, which are more convenient for the practical in-situ application. The synergistic effects of N-CPDs@FLBP and AuNPs can provide a higher conductive interface with the adsorption of rutin by the π–π stacking interaction and cation–π interaction, which are benefit for the improvement of the electrochemical sensing ability to rutin [41, 43]. These interactions lead to the accumulation of rutin on the electrode surface, and the resultant better analytical performances. Finally, the feasibility is verified by comparing the results of real samples of rutin pharmaceutical tablet and Chinese medicine plant FSI on CILE based sensor and SPE based sensor. This study provides a new idea for rapid determination of trace rutin in the Chinese medicine plants, and provides a procedure for the intelligent monitoring the quality of Chinese medicine.

Point 7: Correction on English language is necessary throughout the manuscript.

Response 7: Thanks for the reviewer's comments, English language of the whole manuscript have been corrected to our best.

Reviewer 4 Report

The authors are presenting an electrochemical sensor for rutin. However, I find many important control experiments were not done. Rutin is made up of quercetin and retinose linked with disaccharide bond. The individual effect of only quercetin and retinose along with few other carbohydrates like - glucose, mannose in the experimental set up, need to be shown. It would actually distinguish individuak cointribution from CH-pi or cation-pi or pi-pi stacking interaction.

Additionally, effect of simple catechol or resorcinol types of compound should also be inverstigated in their electrochemical experimental set up.

Otherwise, the paper does not have any insights and scientific merit and should not be accepted.

Round 2

Reviewer 4 Report

It can now be accepted.